# A new mouse mutant with a discrete mutation in *Pcdhgc5* reveals that the Protocadherin γC5 isoform is not essential for dendrite arborization in the cerebral cortex

Camille M. Hanes[1], David M. Steffen[1], George C. Murray[2], Robert W. Burgess[2], Joshua A. Weiner [1], Andrew M. Garrett [3]*

**1** Department of Biology and Iowa Neuroscience Institute, University of Iowa, Iowa City, Iowa, United States of America, **2** The Jackson Laboratory, Bar Harbor, Maine, United States of America, **3** Department of Pharmacology and Department of Ophthalmology Visual and Anatomical Sciences, Wayne State University School of Medicine, Detroit, Michigan, United States of America

* andrew.garrett@wayne.edu

## Abstract

There are ~60 clustered protocadherin (cPcdh) isoforms expressed from three gene clusters (*Pcdha*, *Pcdhb*, *Pcdhg*) arrayed in tandem across nearly 1 Mb in mammals. cPcdhs are homophilic cell adhesion molecules (CAMs) critical for a host of neural developmental functions consistent with a role in cell-cell recognition. Indeed, isoforms make recognition modules in combination to generate recognition diversity far exceeding the ~60 individual CAMs. However, there is also growing evidence for specialized functions for specific isoforms, particularly the C-type isoforms found at the 3' ends of the *Pcdha* cluster (αC1 and αC2) and at the 3' end of the *Pcdhg* cluster (γC3, γC4, and γC5). We have previously described unique roles for γC3 in dendrite arborization in the cerebral cortex and neural circuit formation in the spinal cord, as well as for γC4 in neuronal survival. Here we report a new mouse mutant specifically targeting the *Pcdhgc5* exon encoding γC5. Unlike the rest of the *Pcdhg* cluster, expression of this isoform does not begin until postnatal stages of mouse development, increasing in the second week of life, suggesting specialized roles. We found significant expression changes in gene pathways involved in synaptic activity, learning and memory, and cognition. Despite this, we saw no major disruption in the cerebral cortex in neuronal organization, survival, dendritic arborization, or synaptic protein expression in these mutants. This new model will be an important tool for future studies delineating specific functions for γC5.

## Introduction

The clustered protocadherins (cPcdhs) are a family of ~60 homophilic cadherin superfamily cell adhesion molecules (CAMs) with numerous essential functions in the

**Data availability statement:** All RNA-seq data are available in the GEO database with accession number GSE310953. Other data are included in Supporting information.

**Funding:** NIH/NINDS R21 NS090030 to J.A.W. and R.W.B. NIH/NINDS R01 NS055272 to J.A.W. NIH/NEI R01 EY031690 to A.M.G. The funders played no role in the study design, data collection and analysis, decision to publish, or preparation of the manuscript.

**Competing interests:** The authors have declared that no competing interests exist.

developing nervous system. cPcdhs are expressed from three contiguous gene clusters in mammals: in the mouse, the *Pcdha* cluster encodes 14 α-Pcdhs, the *Pcdhb* cluster encodes 22 β-Pcdhs, and the *Pcdhg* cluster encodes 22 γ-Pcdhs [1,2]. Much of each individual isoform is encoded by a single large variable exon, including six extracellular cadherin (EC) repeats, a transmembrane domain, and a unique variable cytoplasmic domain. The *Pcdha* and *Pcdhg* clusters each have a set of constant exons at their 3' ends that encode a shared cytoplasmic C-terminus specific to that individual cluster, while *Pcdhb* has no such constant exons such that each β-Pcdh is encoded entirely by the single variable exon (Fig 1A). Individual *Pcdhg* isoforms are further classified based on sequence homology as belonging to the γA, γB or γC subgroups (Fig 1B). The *Pcdha* cluster also has two C-type isoforms; altogether, the five C-type isoforms share more homology with each other than they do with the other members of their respective *Pcdha* and *Pcdhg* clusters. Each cPcdh isoform interacts in a strictly homophilic manner in *trans* (across two cells) but each promiscuously can form *cis* dimers with less isoform-specific stringency [3–5]. This organization of protein binding can result in the formation of a large lattice of dimers between membranes with isoform matching, and can thus encode for a very high level of recognition diversity [6–8].

The large number of isoforms encoded by the cPcdhs raises the question of whether there is functional redundancy among the isoforms, or if all 58 are uniquely needed for proper development. Some cell types do require numerous isoforms to provide a mechanism for self/non-self discrimination important for neuronal self-avoidance, without a clear requirement for any one individual isoform [9–11]. However, the expression pattern of the C-type isoforms is more consistent with specialization. Three of the five (αC2, γC4, γC5) lack the conserved sequence element found in the promoters of nearly all other *cPcdh* exons, suggesting these isoforms are regulated in a different manner [1,2]. The C-type isoforms are highly expressed, often deterministically or preferentially in some cell types: αC2 is the dominant isoform in serotonergic neurons [12,13], γC4 in cortical interneurons *vs.* γC5 in cortical excitatory neurons [14], and γC3 in astrocytes of many brain regions [15]. This suggests that the primary role of the C-type isoforms may not be for self/non-self discrimination, as the widespread expression of these isoforms would seem to argue against this.

The expression patterns of C-type isoforms could, rather, suggest distinct roles. Consistent with this, we employed a discrete *Pcdhgc3* mouse mutant to demonstrate unique roles for γC3 in dendritic arborization [16,17], spinal sensory neuron synapse formation [18] and astrocyte morphology [15]. The γC4 isoform is the only one necessary and sufficient for neuronal and organismal survival [16,19–21]. Additionally, αC2 has a unique role in establishing axon morphology in serotonergic neurons [12,13]. We therefore reasoned that the remaining γ-Pcdh C-type isoform, γC5, may also play unique roles in neural development. Indeed, it has been reported that γC5 localizes to GABAergic synapses, where it interacts with the γ2 subunit of the postsynaptic GABA$_A$ receptor (γ2-GABA$_A$R) via the two proteins' cytoplasmic domains [22]. Although γC5 is not required for GABAergic synapse formation, this interaction

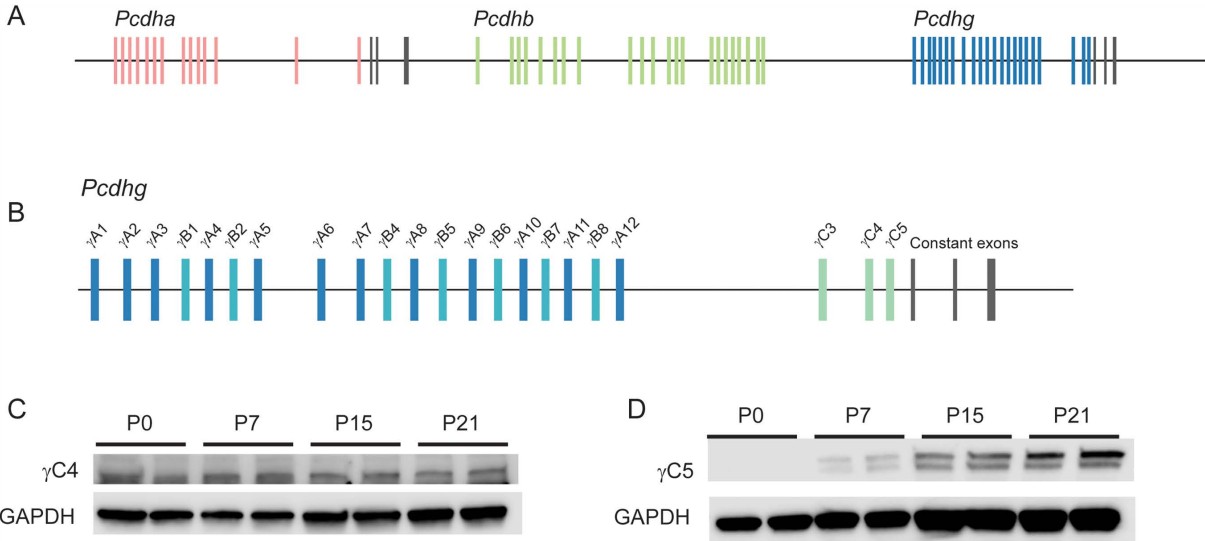

**Fig 1. cPcdh locus and isoform expression. (A)** The cPcdhs comprise three gene clusters: *Pcdha* (red), *Pcdhb* (green), and *Pcdhg* (blue). *Pcdha* and *Pcdhg* each have constant exons at the 3' end of the locus (black). **(B)** Variable exons in the *Pcdhg* cluster are classified as A-, B-, or C-type. γC5 is at the 3' end of the variable exons. **(C)** Western blot analysis of γC4 expression in brain lysates during postnatal development demonstrated stable expression from birth, while **(D)** γC5 was undetectable at P0 and increased from low expression at P7 to high expression at P21. GAPDH was used as a loading control for both isoforms.

facilitates GABA$_A$R cell surface expression and GABAergic synapse stability and maintenance [22]. Similar to γC3 and γC4, γC5 is thought to be widely expressed in the central nervous system; however, the temporal and cell-type expression patterns of γC5 appear distinct from those of other γ-Pcdh isoforms. For example, mouse γC5 expression has been reported to begin only during the second postnatal week [23], coinciding with synaptogenesis [24]. This is in contrast to both γC4, which is highly expressed embryonically and within the first two postnatal weeks before declining, and several other γ-Pcdh isoforms, whose expression is relatively uniform during development and into adulthood [23]. Analysis of single-cell RNAseq data sets from adult mouse cortex observed a contrasting pattern of γC4 and γC5: γC4 is distinctly enriched in GABAergic interneurons compared to excitatory neurons, whereas glutamatergic excitatory neurons are enriched in γC5 expression [14,25].

We thus hypothesized that γC5 plays a unique role in the central nervous system. Here, we test this hypothesis with a new targeted mouse line harboring a discrete mutation disrupting *Pcdhgc5*. Despite significant expression changes in transcriptional pathways involved in synaptic activity, learning and memory, and cognition, we find no gross disruption in the morphology of the brain, nor in the organization, survival, dendritic arborization, or synaptic protein expression of cerebral cortex neurons in mice lacking γC5. While distinct roles for γC5 remain to be uncovered by further analysis, this result further highlights the uniqueness of demonstrated γC3 and γC4 functions and confirms that cPcdh C-type isoforms are not interchangeable.

## Materials and methods

### Mouse lines

All experiments included male and female mice and were carried out in accordance with the Institutional Animal Care and Use Committee at the University of Iowa, the Jackson Laboratory, and Wayne State University, as well as the National Institutes of Health guidelines. All mice were kept under standard housing and husbandry conditions with food and water provided *ad libitum* and 12/12 hour light/dark cycles. All control and mutant mice were on a C57BL/6J background. Control

mice for all mutants were either wild-type littermates or age-matched C57BL/6J mice from the same colony and housed together. The *Thy1-YFPH* line [26] used for dendrite arborization and dendritic spine analysis was obtained from The Jackson Laboratory (Strain 003782).

### *PcdhgC5KO* allele

*Pcdhg$^{C5KO}$* mice (*γC5-KO*) were generated by specifically targeting the *Pcdhgc5* variable exon using CRISPR/Cas9 genome editing. An sgRNA (5'-TGGCATCACCACAGGTCGCT<u>GGG</u>-3'; PAM site underlined) complementary to the variable exon of C5 was microinjected (50 ng/µL), along with Cas9 mRNA (100 ng/µL), into C57BL/6J zygotes, which were subsequently implanted into pseudopregnant female C57BL/6J mice. Founders were identified by PCR genotyping, and one founder carrying the 7-base pair (bp) deletion of TCGCTGG, 25 bp after the start codon for *PcdhgC5* exon 1, was bred to C57BL/6J mice for two generations to eliminate mosaicism and possible off target effects. Mice were genotyped via PCR using primers spanning the CRISPR target site (C5KO forward: 5'-GGCAAACCTCAGAGCAGTTT-3' and C5KO reverse: 5'- GACAACTCTCCCAGCTTTGC-3').

### Tissue preparation and immunofluorescence

Experimental and control mice at 6 weeks of age were transcardially perfused with ice cold 4% paraformaldehyde (PFA) and brains were extracted and immersed overnight in the same fixative at 4°C. For cryostat sectioning, fixed tissues were sunk in 30% sucrose in PBS at 4°C, immersed in Tissue-Tek OCT compound, and frozen in an isopentane-cooled dry ice/ethanol bath. Cryostat sections (collected using a Leica CM1850 cryostat) were cut at 18 µm and thawed onto gelatin-subbed slides. For immunofluorescence, sections were rehydrated, blocked with 2.5% bovine serum albumin (BSA; Sigma, Fraction V), 0.1% Triton X-100 in PBS, and incubated with primary antibody overnight at 4°C in the same solution in a humidified chamber. The next day, sections were washed with PBS and incubated with secondary antibodies conjugated with Alexa-488, −568, or −647 (1:500, Invitrogen) for 1 hour at room temperature (RT). Slides were then washed and DAPI was added to the final wash to stain nuclei. FluoroGel (Electron Microscopy Sciences) was used to mount coverslips over the sections.

### Vibratome sectioning

Brains were fixed as above, then embedded in 2% agarose and sectioned coronally at 200 µm using a Vibratome 3000 sectioning system. Brain sections were then mounted onto slides with Fluoro-Gel for confocal microscopy analysis.

### Image collection for dendritic arborization analysis

Images were collected from pyramidal neurons in the S1 cortex using a 20x objective on a Leica TCS SPE confocal microscope via the Leica LAS X software. Stacks were 100 µm in the Z-dimension with a step size of 0.5 µm. Individual neurons were traced in 3D using FIJI's Simple Neurite Tracer plugin. Sholl analysis was conducted for each individual neuron using FIJI's Sholl Analysis plugin. Area under the curve for each neuron was calculated using Prism (GraphPad). Researchers were blind to genotype during imaging and analysis.

### SDS/PAGE and Western blotting

Brain lysate was prepared by homogenizing brains in 1−2 ml RIPA buffer (50 mM Tris-HCl pH 7.4, 5mM NaF, 0.1% SDS, 0.25% sodium deoxycholate, 1% NP-40, 0.15M NaCl) supplemented with protease inhibitor (Roche miniComplete) and PhosSTOP phosphatase inhibitor (Roche). Protein samples of equal amounts were loaded into TGX precast gels (Bio-Rad), separated via SDS/PAGE, and transferred onto nitrocellulose membranes using a TransBlot Turbo System (Bio-Rad). After protein transfer, membranes were blocked in 5% nonfat milk or BSA in TBS with 0.1% Tween 20 (TBST) for

1 h. Membranes were then washed 3x for 5 min in TBST. Primary antibodies were diluted in 2.5% BSA in TBST and membranes were incubated overnight at 4°C. Blots were washed 3x in TBST. The membrane was then incubated for 1 h in horseradish peroxidase (HRP)-conjugated secondary antibodies (1:1000–1:5000, Jackson ImmunoResearch) diluted in 2.5% BSA in TBST. Blots were washed 3x in TBST with a final wash in deionized water. Signals were detected using SuperSignal West Pico or Femto Enhanced Chemiluminescent Substrates (Thermo Fisher Scientific) on a LI-COR Odyssey FC imager.

GAPDH, detected by reprobing blots with a specific primary antibody, served as the loading control unless noted. In some experiments, Revert™ 700 Total Protein Stain (LI-COR) was used instead to ensure equal loading. Briefly, immediately after transfer, nitrocellulose membranes were incubated in Revert™ 700 staining solution for one minute and then rinsed with a wash solution. Membranes were imaged in the 700nm channel of a LI-COR Odyssey FC imager with an exposure of one minute. Membranes were incubated in a destaining solution for 5–10 minutes before proceeding to the blocking step.

### Synaptosome preparations

Cortical tissue was homogenized in Syn-PER synaptic protein extraction buffer (ThermoScientific) at 10 ml of buffer per gram of tissue. All centrifugation occurred at 4° C. Lysate was first cleared by centrifuging at 1,200 x g for 10 mins. The pellet was discarded and the resulting supernatant (total homogenate) was transferred to a new tube. Approximately 200 µL of this supernatant was saved; the remaining supernatant was centrifuged at 15,000 x g for 20 mins. The resulting supernatant (cytosolic fraction) was transferred to a new tube and saved. The synaptic pellet was resuspended in ~400 µL of Syn-PER™ buffer.

### Antibodies

The following primary antibodies were used at the indicated concentrations.

Pan-γ-Pcdh N159/5 (Mouse, 1:500, NeuroMab 75−185, RRID:AB_2877195); Pan-γ-Pcdh-A N144/32 (Mouse, 1:500, NeuroMab 75−178 RRID:AB_2877459); γ-Pcdh-B2 N148/30 (Mouse, 1:500, NeuroMab 75−184, RRID:AB_2877457); γ-Pcdh-C5 (Mouse, 1:500, Invitrogen MA5−26922 RRID:AB_2724958); GAPDH (Mouse, 1:1000, Abcam 8245, RRID:AB_2107448); CC3 (Rabbit, 1:100, Cell Signaling 9661, RRID:AB_2341188); GFAP (Mouse, 1:500, Sigma-Aldrich G3893, RRID:AB_477010); NeuN (Mouse, 1:300, Millipore MAB377 RRID:AB_2298772); Cux1 (Rabbit, 1:100, Proteintech 11733–1-AP RRID:AB_2086995); Foxp2 (Rabbit, 1:500, Abcam 16046 RRID:AB_2107107); PSD95 (Mouse, 1:1000, Cell Signaling 36233 RRID:AB_2721262);

Gephyrin (Rabbit, 1:1000, Cell Signaling 14304, RRID:AB_2798443); Synapsin (Rabbit, 1:1000, Cell Signaling 5297 RRID:AB_2616578).

### Quantitative RT-PCR (qPCR)

Whole brains were collected from mice at postnatal day (P)21 and RNA isolated using TRIzol reagent (ThermoFisher) according to the manufacturer's protocol. RNA was further purified using the RNeasy Midi cleanup kit (Qiagen) according to the manufacturer's protocol. Two µg of total RNA was used for cDNA synthesis using High-Capacity cDNA Reverse Transcription kit (ThermoFisher). The cDNA produced was diluted 1:2 for use with LightCycler 480 SYBR Green I Master (Roche) with primers for Pcdhgc5 (GTTCCCGCTCTAGTACGCTG and CAGGTGCCAGTTTCATCACC) and for GAPDH (AATGTGTCCGTCGTGGATCT and GTTGAAGTCGCAGGAGACAA). The QuantStudio 3 Real Time PCR system (ThermoFisher) was used to carry out qPCR reactions. Relative abundance of the target gene was calculated using the ΔΔCt method, normalized to GAPDH within each sample, with mutant mice then compared to control mice.

## RNAseq

Cerebral cortex from 3 male and 3 female homozygous *γC5-KO* mice and 5 male and 1 female wild type were collected and snap frozen in liquid nitrogen before storage at -80C. Total RNA was isolated from tissue using the NucleoMag RNA Kit (Macherey-Nagel) and the KingFisher Flex purification system (ThermoFisher). Tissues were homogenized in MR1 buffer (Macherey-Nagel) using a gentleMACS dissociator (Miltenyi Biotec Inc). RNA isolation was performed according to the manufacturer's protocol. RNA concentration and quality were assessed using the Nanodrop 8000 spectrophotometer (Thermo Scientific) and the RNA ScreenTape Assay (Agilent Technologies). Libraries were constructed using the KAPA mRNA HyperPrep Kit (Roche Sequencing and Life Science), according to the manufacturer's protocol. The quality and concentration of the libraries were assessed using the D5000 ScreenTape (Agilent Technologies) and Qubit dsDNA HS Assay (ThermoFisher), respectively. Libraries were sequenced 150 bp paired-end on an Illumina NovaSeq 6000 using the S4 Reagent Kit v1.5. Analysis of RNAseq data in fastq format was performed using an open source Nextflow pipeline (v3.5) at The Jackson Laboratory comprised of tools to perform read quality assessment, alignment, and quantification [27]. The pipeline takes as input fastq files for a single sample and outputs read alignment counts. FastQC (0.11.9) was used for quality checks and Trim Galore (0.6.7) was used to remove adapters and sequences with low quality (Fred<20). Sequence reads that passed quality control were aligned to mouse reference (GRCm38) using STAR (v2.7) and gene expression estimates were made using RSEM (v1.3) with default parameters. Differential expression analysis was performed using a differential expression script in R developed by Computational Sciences at The Jackson Laboratory based on the EdgeR package [28]. We selected FDR < 0.05 and absolute $Log_2FC > 0.5$ as cutoffs for differential expression.

## Results

### Generation of the γC5-KO mouse

Expression of protocadherin γC5 has been reported to begin postnatally, in contrast to the other 21 members of the family [23]. We verified this by Western blot on brain lysates with isoform-specific antibodies (Fig 1). While γC4 expression was already strong at birth with stable expression through the first three weeks of life, γC5 was undetectable at P0 with increasing expression from P7 to P21 (Fig 1C and 1D). This suggests a specialized role for γC5 in later developmental processes. To probe such roles, we created a discrete γC5 knock-out mouse by targeting the 5' end of the *Pcdhgc5* variable exon using CRISPR/Cas9 genome editing. The resulting mutation generated a 7-base pair (bp) deletion 25 bp downstream of the start codon, creating a frameshift resulting in an early stop codon (Fig 2A). The new strain was dubbed *Pcdhg^C5KO*, referred to here as γC5-KO. Homozygous mutants proved to be viable and fertile with no obvious gross abnormalities.

Quantitative PCR showed a significant reduction in *Pcdhgc5* transcripts in mutant brains, as expected: the frame-shift mutation should not completely abrogate transcription but is likely to induce some degree of nonsense-mediated decay (Fig 2B). Complete loss of γC5 protein in homozygous mutants, across both cytoplasmic and synaptic compartments, was confirmed by Western blotting (Fig 2C) as well as by immunostaining of the cerebral cortex with a γC5-specific antibody (Fig 2D). To test if disruption of the *Pcdhgc5* variable exon affected expression across the gene cluster, we quantified levels of other cPcdh isoforms in mutant brains utilizing available monoclonal antibodies detecting 11 of the 12 γ-Pcdh A isoforms, γB2 specifically, or the constant domain of all α-Pcdhs (Fig 2E; [29]). Interestingly, we found that the γ-Pcdh A protein levels were reduced to ~70% of controls in γC5 KOs, while γB2 and the α-Pcdhs were unaltered (Fig 2E). As γC5 expression increased postnatally (Fig 1D), we tested the extent to which total γ-Pcdh levels would be reduced in the cortex of *γC5-KO* animals compared with controls over time. We performed Western blots of brain lysates at P7, P15, and P21 and observed more clear reduction of total γ-Pcdh protein in older animals (Fig 2F, total protein stain used as loading control, not shown). This may be due in part to reduced A-type isoforms, but the pattern towards greater reduction in older mice likely indicates that γC5 makes up a relatively large fraction of the γ-Pcdh pool at the later stages of development, particularly in cortical pyramidal neurons [14].

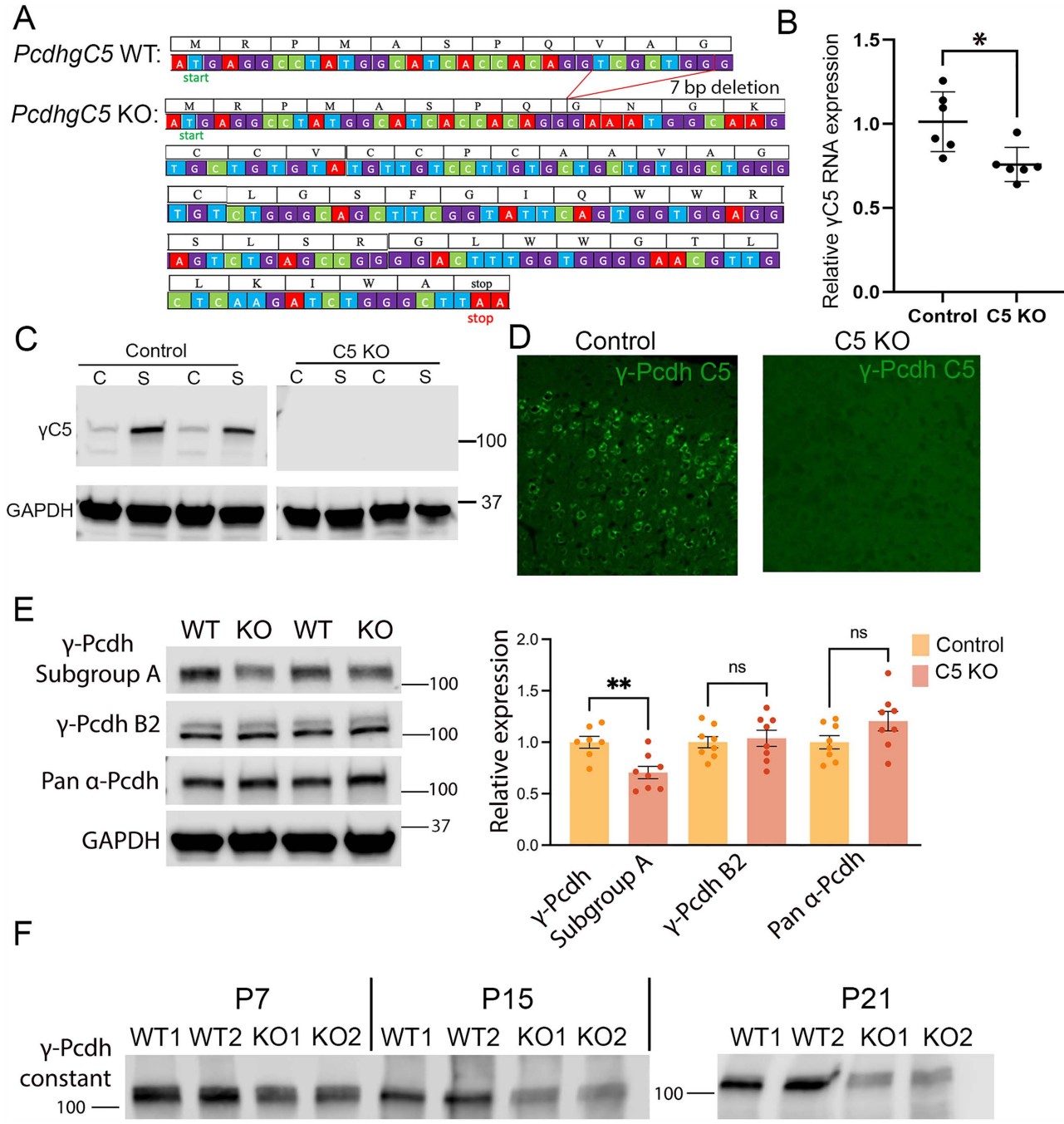

**Fig 2. Generation of a γC5 knockout mouse. (A)** Start of the mouse *PcdhgC5* gene with protein translation. A seven base-pair (bp) deletion in the *PcdhgC5* variable exon results in a frameshift, leading to an early stop codon. **(B)** Quantitative PCR performed on six-week control and γC5 KO mice show a significant nonsense-mediated reduction of *PcdhgC5* transcripts in γC5 KO compared to control (p = .012). A forward primer in the 3' end of the *PcdhgC5* variable exon and a reverse primer in the first *Pcdhg* constant exon were used. Note that the forward primer does not span the 7 bp deletion; thus, we expect to see some *PcdhgC5* transcription. However, there is no translation of full-length γC5 in the γC5 KO mutant, evident by the lack of γC5 signal in Western blots (C) and immunostaining (D). n = 6 mice per genotype run in triplicate. **(C)** Western blots from cytoplasmic ("C") and synaptic ("S") fractions of cortical brain lysate from control and γC5 KO mice demonstrate the synaptic enrichment of γC5 protein in control and the absence of all γC5 protein in γC5 KO mice. GAPDH serves as the loading control. n = 2 mice per genotype. **(D)** Immunostaining of control and γC5 KO cortical Layer 2/3 at six weeks of age using a γC5-specific antibody demonstrates the absence of γC5 expression in γC5 KO mice. **(E)** Western blots of cortical lysate from P60 control and γC5 KO mice reveal a significant decrease in γ-Pcdh subgroup A isoforms in γC5 KO cortex (p = .0041), while γ-Pcdh B2 and

pan-α-Pcdh levels are unaltered. n = 7-8 mice per genotype run in duplicate. Unpaired t-test; error bars represent the SEM; ns = not significant. **(F)** Western blots from wildtype and γC5 KO mice at various postnatal ages using a pan-γ-Pcdh antibody. Prior to the onset of γC5 expression (P7), there is little difference between pan-γ-Pcdh expression in wildtype and γC5 KO. However, at ages following the onset of γC5 expression (P15 and P21) a striking decrease in pan-γ-Pcdh expression is observed in γC5 KO mice. This decrease in pan-γ-Pcdh expression likely reflects the loss of γC5, being a highly expressed C-type isoform, as well as the significant decrease in γ-Pcdh Subgroup A isoforms observed in C5 KO animals. Revert™ Total Protein Stain was performed on the blot prior to antibody incubation to ensure equal protein loading (not shown). n = 2 mice per genotype per age.

We next examined γC5-KO cerebral cortex for any major disruptions to neuronal survival or cell organization. We stained cryosections from mutants and controls with markers for all neurons (NeuN, Fig 3A and 3B), deep layer neurons (Foxp2, Fig 3C and 3D), and upper layer neurons (Cux1, Fig 3E and 3F). We observed no defects in gross brain morphology or neuronal layering. The γC4 isoform has been shown to uniquely control neuronal survival [19–21], including that of cortical interneurons [14,30,31]. Consistent with this specialized role for γC4, we did not observe any increase in the apoptotic marker cleaved caspase-3 (Fig 3I and 3J) nor in activated astrocytes (GFAP, Fig 3G and 3H) in γC5-KO mutants.

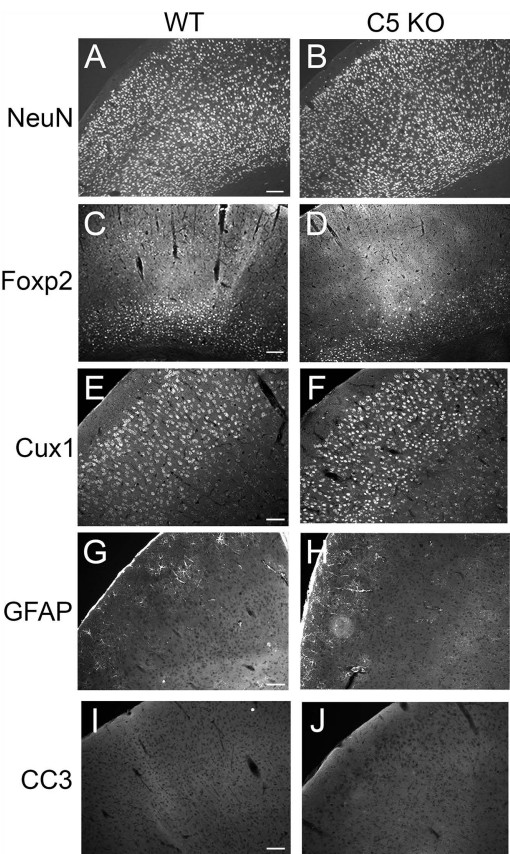

**Fig 3. γC5 KO brains have grossly normal morphology.** Cryosections of adult control and γC5 KO cortices stained with the indicated layer- and cell type-specific markers (A-B, NeuN, neurons; C-D, FoxP2, deep layer neurons; E-F, Cux1, upper layer neurons; G-H, GFAP, astrocytes; I-J, cleaved caspase-3 [CC3], apoptotic cells), revealing grossly normal morphology and neuronal survival in γC5 KO mutants. Scale bar for NeuN and Foxp2: 100 μm. Scale bar for Cux1, GFAP, and CC3: 50 μm.

## Synaptic protein levels are unperturbed in adult γC5-KO cortex

The γ-Pcdh protein family has been implicated in the control of synapse development [18,32–35]. Additionally, due to the reported dichotomous expression of γC5 in excitatory neurons and γC4 in inhibitory neurons [14], as well as the reported interaction of γC5 and the γ2 subunit of GABA$_A$ receptors [22], we reasoned that both excitatory and inhibitory synapses could be altered in γ*C5-KO* mice. Western blot analysis was performed using P60 control and γ*C5-KO* cortical lysates, using specific antibodies to quantify levels of PSD95, an excitatory postsynaptic protein, gephyrin, an inhibitory postsynaptic protein, and synapsin, a protein implicated in the regulation of neurotransmitter release. No statistical difference in levels of any of these proteins was observed between control and γ*C5-KO* mice (Fig 4A and 4B), suggesting—along with the lack of any outward signs of aberrant behavior--that synapses are not grossly altered in the absence of the γC5 isoform.

## Dendritic arborization is not altered in γC5-KO mice

We have previously demonstrated critical roles for the γ-Pcdhs in cortical dendritic arborization, including mechanisms requiring the constant domain common to all isoforms [36–38], as well as an isoform-specific role for the variable cytoplasmic domain of γC3 [16,17]. If any other γ-Pcdh isoform were likely to influence dendritic arborization specifically, it would be γC5, given its predominant expression in cortical pyramidal neurons and its increase in expression during a time of extensive dendrite growth and branching. Therefore, we crossed γ*C5-KO* mice with those harboring the *Thy1-YFPH* allele to label a subset of cortical pyramidal neurons [26]. Mice at six weeks of age were collected and Sholl analyses were performed on layer 5 neurons to assess dendritic complexity. Despite the broad expression of γC5 in excitatory neurons of the cortex, we found no significant alterations in dendritic arborization in γC5 KOs compared to controls (Fig 5A–5C). Because single-cell RNAseq data sets indicate that *Pcdhgc5* transcription is higher in layer 2/3 pyramidal neurons compared to those in layer 5 [39], we reasoned that if γC5 *does* affect branching even slightly, we might be more likely to observe an effect of its loss in these upper-layer neurons, a subset of which is also labeled by the *Thy1-YFPH* transgene. Sholl analysis again did not reveal any significant differences in dendritic complexity between γ*C5-KO* and control neurons in layer 2/3 (Fig 5D–5F). Thus, γC5 is not required for normal dendritic arbor development in the cerebral cortex.

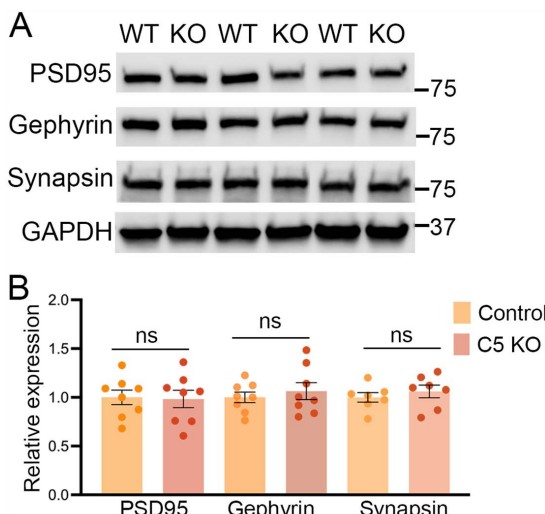

**Fig 4. Synaptic protein levels are unperturbed in γC5 KO adult cortex. (A)** Cortical lysate from P60 control and γC5 KO were probed with synaptic proteins (PSD95, gephyrin, and synapsin) via Western blotting. **(B)** No significant differences were observed between control and γC5 KO. n = 8 mice/genotype, run in duplicate. Each dot represents one mouse averaged from at least 2 separate Western blot experiments. Unpaired t-test; error bars represent the SEM; ns = not significant.

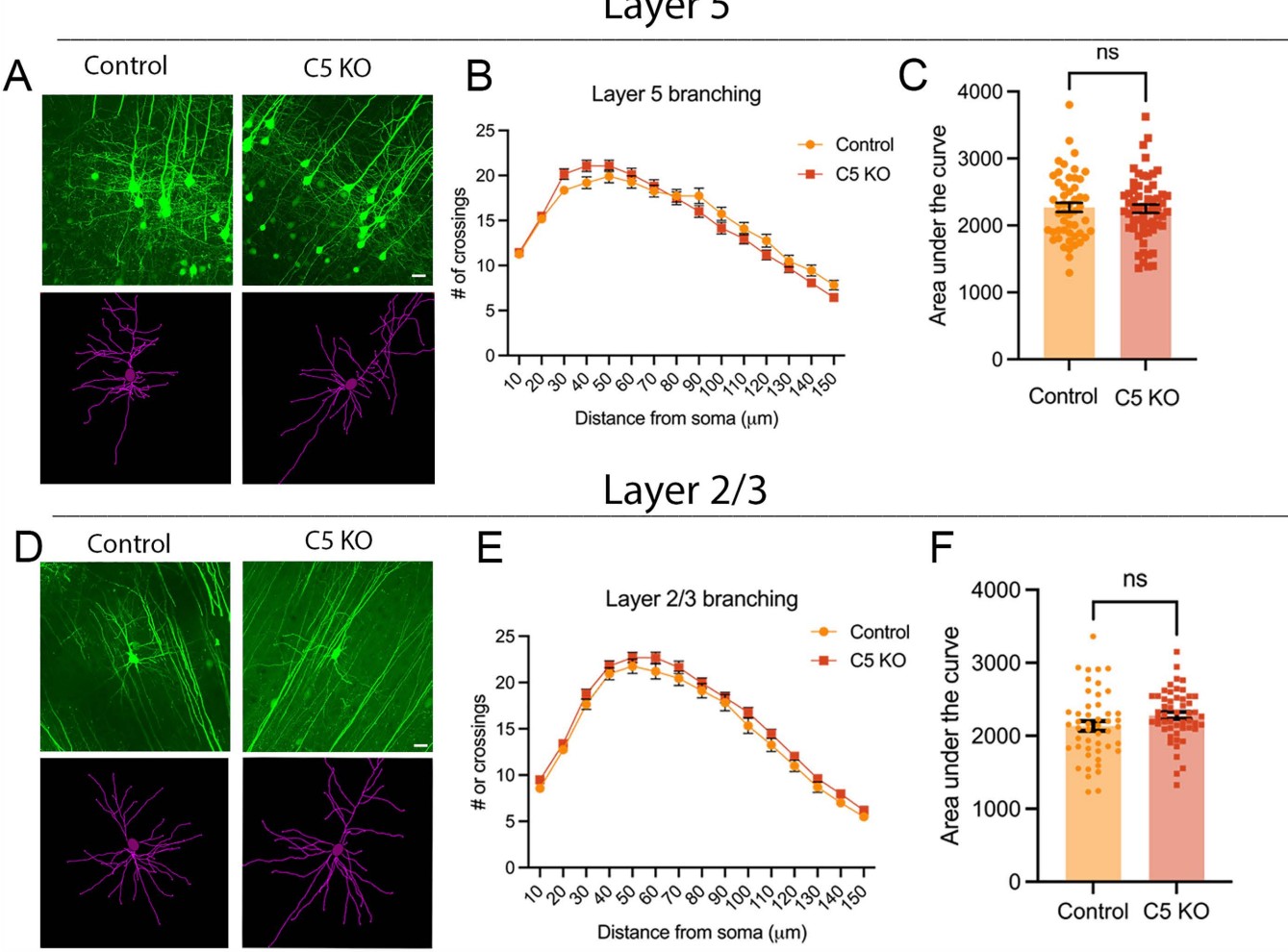

**Fig 5. Dendritic arborization is not perturbed in Layer 5 or Layer 2/3 in γC5 KO mice.** Top panel: **(A)** Representative YFP images (top) and tracings (bottom) of Thy1-YFPH-labeled Layer 5 pyramidal neurons of mice six weeks of age. Scale bar: 25 μm. **(B)** Sholl analysis graphs show dendritic crossings at spheres of increasing 10-μm intervals of mice at six weeks of age. **(C)** Graph showing area under the curve of Sholl graph. Bottom panel: **(D)** Representative YFP images (top) and tracings (bottom) of Thy1-YFPH-labeled Layer 2/3 pyramidal neurons of mice six weeks of age. **(E)** Sholl analysis graphs show dendritic crossings at spheres of increasing 10-μm intervals of mice at six weeks of age. **(F)** Graph showing area under the curve of Sholl graph. No significant differences in dendritic complexity are seen in γC5 KO compared to control in either Layer 5 or Layer 2/3. n = ≥ 51 neurons across 4-6 mice (Layer 5); ≥48 neurons across 3-6 mice (Layer 2/3). Unpaired t-test; error bars represent the SEM; ns = not significant.

## RNAseq analysis reveals differentially expressed genes in γC5-KO cortex

Given the lack of clear structural or molecular phenotypes in γ*C5-KO* brains, we turned to transcriptomic analyses to gain insight on potential functions for γC5. Cerebral cortices were isolated from male and female control (n = 6) and γ*C5-KO* (n = 6) mice at 4 weeks of age for RNAseq analysis. First, we analyzed the expression of cPcdh isoforms for potential off-target disruptions (Table 1). In confirmation of our quantitative RT-PCR analysis, *Pcdhgc5* itself was significantly reduced (Log$_2$FC = −0.389, FDR = 2.90 X10$^{-4}$). Despite our identification of reduced γA expression at the protein level, the only change in any other cPcdh isoforms detectable at the transcript level was a significant increase in *Pcdhgc4* (Log$_2$FC = 1.205, FDR = 2.27X10$^{-40}$), encoded in part by the variable exon adjacent to that of *Pcdhgc5* (Fig 1B).

**Table 1. Relative expression of cPcdh isoform transcripts in γC5 KO compared to control cortex.**

| Pcdha | logFC | FDR | Pcdhb | logFC | FDR | Pcdhg | logFC | FDR |
|---|---|---|---|---|---|---|---|---|
| α1 | −0.083 | 1 | β2 | 0.027 | 1 | γA1 | −0.051 | 1 |
| α2 | 0.214 | 0.52 | β3 | 0.037 | 1 | γA2 | 0.019 | 1 |
| α3 | 0.050 | 1 | β4 | 0.118 | 1 | γA3 | 0.038 | 1 |
| α4 | 0.035 | 1 | β5 | 0.028 | 1 | γB1 | 0.147 | 0.95 |
| α5 | −0.127 | 1 | β6 | −0.119 | 1 | γA4 | 0.027 | 1 |
| α6 | −0.188 | 0.63 | β7 | −0.006 | 1 | γB2 | 0.000 | 1 |
| α7 | −0.058 | 1 | β8 | 0.234 | 0.57 | γA5 | 0.081 | 1 |
| α8 | 0.059 | 1 | β9 | 0.133 | 1 | γA6 | 0.038 | 1 |
| α9 | 0.089 | 1 | β10 | −0.065 | 1 | γA7 | −0.142 | 0.96 |
| α10 | 0.190 | 0.69 | β11 | 0.084 | 1 | γB4 | −0.116 | 1 |
| α11 | 0.042 | 1 | β12 | −0.022 | 1 | γA8 | −0.053 | 1 |
| α12 | 0.015 | 1 | β13 | 0.048 | 1 | γB5 | 0.044 | 1 |
| αc1 | 0.157 | 0.90 | β14 | −0.017 | 1 | γA9 | −0.121 | 1 |
| αc2 | 0.011 | 1 | β15 | 0.069 | 1 | γB6 | −0.101 | 1 |
| | | | β16 | −0.011 | 1 | γA10 | −0.032 | 1 |
| | | | β17 | 0.067 | 1 | γB7 | 0.024 | 1 |
| | | | β18 | 0.161 | 0.87 | γA11 | −0.052 | 1 |
| | | | β19 | 0.072 | 1 | γB8 | −0.097 | 1 |
| | | | β20 | 0.165 | 0.79 | γA12 | −0.004 | 1 |
| | | | β21 | 0.013 | 1 | γC3 | 0.020 | 1 |
| | | | β22 | 0.043 | 1 | γC4 | 1.205 | 2.27E-40 |
| | | | | | | γC5 | −0.389 | 0.00029 |

We detected a total of 51 upregulated and 146 down-regulated genes using modest cutoffs of $Log_2FC > 0.5$ and FDR < 0.05. Among the significantly upregulated genes in γC5-KO cortex was the transcription factor *Npas4* (Neuronal PAS domain protein 4) (Fig 6A). *Npas4* is known to control the expression of genes involved in regulating excitatory-inhibitory balance within neural circuits (reviewed by [40]). For example, *Npas4* promotes an increased number of inhibitory synapses onto excitatory neurons and, conversely, increased excitatory input onto inhibitory somatostatin+ interneurons, resulting in enhanced feedback inhibition [40]. Interestingly, RNAseq analyses on mouse medial prefrontal cortex after shRNA-mediated knockdown of *Npas4* revealed differentially regulated genes pertaining to glutamatergic synapses, including a significant upregulation of *Pcdhgc5* [41]. *Npas4* also plays a role in circadian behavior and the suprachiasmatic nucleus's transcriptional response to light [42]. Another circadian-related gene, *Ciart* (circadian associated repressor of transcription), was also upregulated in γC5-KO mice, and our Gene Ontology (GO) enrichment analysis reports rhythmic behavior and circadian rhythm as potential altered pathways in γC5-KO mice (Fig 6B).

Among the significantly downregulated genes in γC5-KO mice was *Lhx8* (LIM homeobox protein 8), a transcription factor that is expressed in postmitotic neurons in the developing medial ganglionic eminence (MGE), and that is thought to play a role in specifying MGE cell fate via modulation of other transcription factors [43]. Interestingly, *Sp8* was also downregulated in γC5-KO (Fig 6A). *Sp8* is a transcription factor that regulates the migration of MGE-derived cortical interneurons [44]. Our GO enrichment analysis reveals terms such as GABAergic synaptic transmission (Fig 6B); thus, future studies investigating these processes may be valuable.

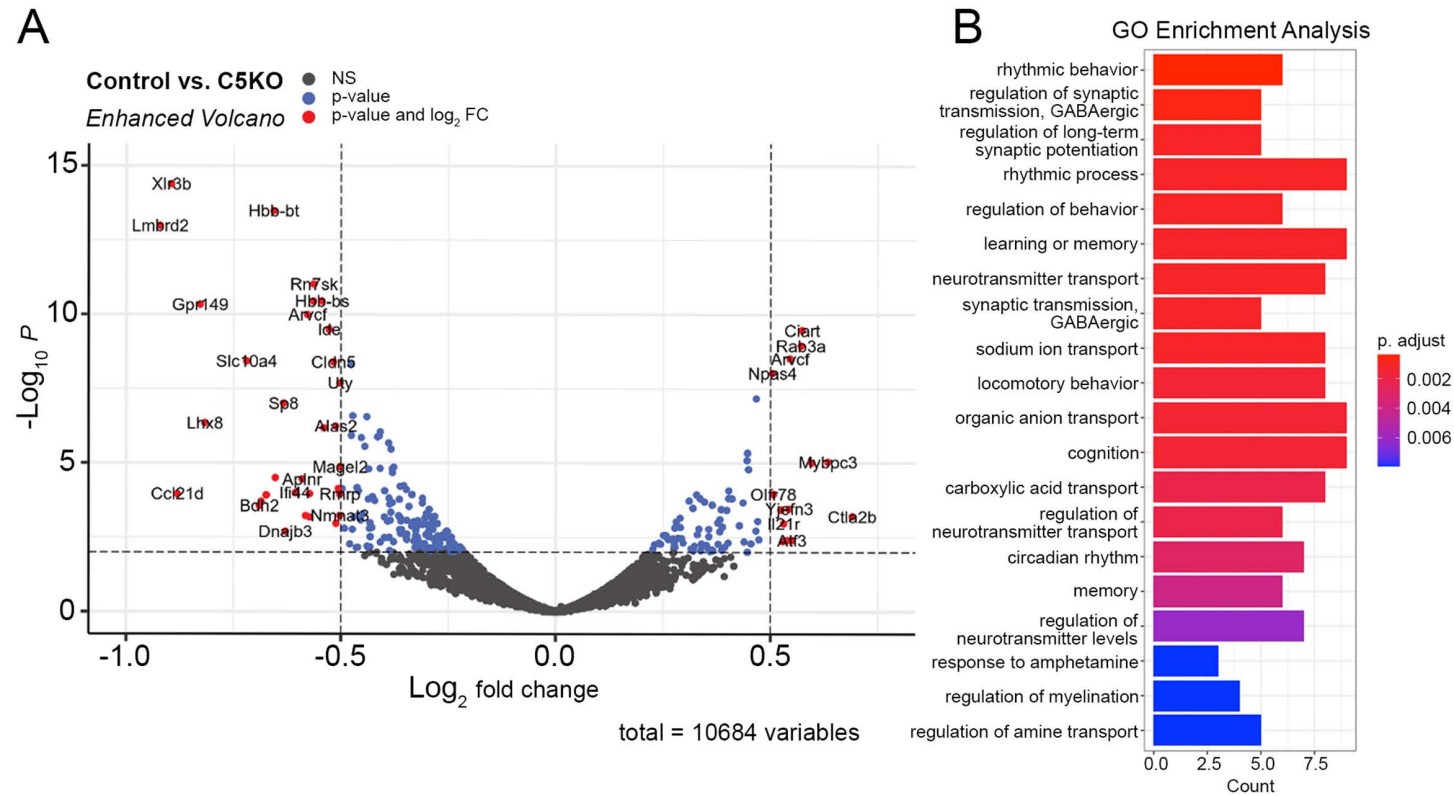

**Fig 6. RNAseq analysis of control and γC5 KO mice. (A)** Volcano plot depicting the statistical significance (y-axis, -log₁₀ p-value) vs. the magnitude of change (x-axis, log₂ fold change) of differentially expressed genes (DEGs) obtained through RNAseq analysis of control and γC5 KO cortex at 4 weeks of age. Median and median absolute deviation of the log₂ fold change were calculated to remove outliers. Outliers that were greater than 20x the median absolute deviation were removed. Long noncoding RNAs and other non-coding RNAs were also filtered out. DEGs were then identified by the criteria of having a false discovery rate (FDR) of <.01 and a fold change cutoff of +/- 0.5. Blue dots represent genes that had a p-value of <.05. Red dots represent genes that met both the p-value cutoff (< 0.05) and the fold change cutoff (+/- 0.5). Black dots represent non-DEGs. **(B)** GO enrichment analysis was performed using the list of DEGs that met both the FDR and fold change cutoff.

## Discussion

Here, we describe a new mouse model with a targeted mutation in the *Pcdhgc5* gene that specifically disrupts the proto-cadherin γC5 protein while leaving the other 21 γ-Pcdhs intact. Having previously described unique roles for γC3 in circuit formation [16,18] and γC4 in neuronal survival [14,19,20], we generated this new model to aid in identifying any unique functions for γC5. In this initial analysis, we found that γC5-KO mice lacked any gross morphological or overt behavioral phenotypes, and exhibited no disruption in neuronal survival, the organization of the cerebral cortex (Fig 3), the levels of pre- and post-synaptic proteins (Fig 4), or dendritic arborization of cortical neurons (Fig 5). Nevertheless, transcriptomic analysis identified that genes in biological pathways including rhythmic behavior, synaptic transmission, and learning and memory were significantly up- or down-regulated in the absence of γC5 (Fig 6). This suggests that further analyses utilizing this new genetic tool may uncover neurobiological functions of this γ-Pcdh isoform, which is uniquely expressed only during the postnatal period.

We previously found that disruption of the entire *Pcdhg* locus in excitatory forebrain neurons resulted in a decrease of ~40% in dendritic complexity compared to controls [36], while discrete loss of only the γC3 isoform yielded nearly the same result: a decrease of ~30% in dendritic complexity compared to controls [16]. Considering the large role γC3 clearly

plays in dendritic arborization, in addition to the role of the γ-Pcdh constant domain via FAK/PKC/MARCKS signaling [37,38], it is plausible that other isoforms and mechanisms simply do not play a large role in dendritic arborization. Consistent with this, the *Pcdhg¹³ᴿ¹* mutant mouse line, which harbors mutations in 9 of the *Pcdhg* genes (but retaining an intact *Pcdhgc3* gene as well as the *Pcdhgc5* gene; [19]), did not exhibit a significant disruption of dendritic complexity [16].

While we were in the course of characterizing our new mutant line, another group reported a different mouse model targeting *Pcdhgc5* with significantly different results from ours [45]. Su et al. described ~25–50% reduction in synaptic proteins PSD95, gephyrin, and synapsin in the cerebral cortices of their mutants, while we found no changes in our homozygous mutants. These authors also described a striking reduction in dendrite arborization in cortical neurons cultured from their mutant, while we found no change *in vivo* [45]. The authors did not provide any characterization of the actual mutation that they generated, but their guide sequences suggest that their goal was to remove the entire *Pcdhgc5* exon (>2000 bp). Conversely, we created a small 7 bp frameshifting deletion that we documented by sequencing, qPCR, Western blot, and immunostaining. The generation of Su et al.'s distinct γC5 knockout mice may have produced off-target effects disrupting other isoforms or reducing their expression. Indeed, in the many previous *Pcdhg* genetic models we have generated and characterized, we found that larger deletions within the locus can significantly reduce expression from otherwise intact variable exons [19,35]. Even with our small mutation, we found minor disruptions in γA protein isoforms here as well as an increase in *Pcdhgc4* transcription (Fig 2 and Table 1). Su et al. (2024) did not report any analysis of other γ-Pcdhs at the mRNA or protein level in their distinct γC5 mutant [45]. Given the confirmed complete loss of γC5 protein in the *γC5-KO* mutant described here (Fig 2), we posit that our findings accurately reflect the isoform-specific importance of γC5 to these processes.

Finally, our RNAseq analyses on cerebral cortex identified differentially expressed genes with gene ontology enriched for GABAergic categories, rhythmic behavior, and circadian rhythms, among others. This could reflect γC5 functions in regulating inhibitory synapses, as has been previously reported [22]. Additional behavioral and physiological analyses will be needed to connect the transcriptional differences we observed to functional outcomes. In addition, transcriptome profiling at earlier time points may help clarify the developmental basis for gene expression changes observed in four-week-old mice analyzed here and help distinguish cause-and-effect between the changes in physiology and gene expression. This new mutant will be essential for those efforts.

## Supporting information

**S1 Raw Images. Full Western blot images.**
(PDF)

**S1 File. Raw data points used for quantification.**
(PDF)

## Author contributions

**Conceptualization:** Camille M. Hanes, David M. Steffen, Robert W. Burgess, Joshua A. Weiner, Andrew Garrett.

**Data curation:** Camille M. Hanes, David M. Steffen, George C. Murray.

**Formal analysis:** Camille M. Hanes.

**Funding acquisition:** Robert W. Burgess, Joshua A. Weiner, Andrew Garrett.

**Investigation:** Camille M. Hanes.

**Methodology:** Camille M. Hanes, David M. Steffen, George C. Murray.

**Project administration:** Robert W. Burgess, Joshua A. Weiner, Andrew Garrett.

**Writing – original draft:** Camille M. Hanes, Joshua A. Weiner, Andrew Garrett.

**Writing – review & editing:** Camille M. Hanes, David M. Steffen, George C. Murray, Robert W. Burgess, Joshua A. Weiner, Andrew Garrett.

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
