## [Decision Letter · Decision Letter 0]

12 Dec 2025

Dear Dr. Garrett,

We look forward to receiving your revised manuscript.

Kind regards,

Xiangming Zha, Ph.D.

Academic Editor

PLOS One

Journal Requirements:

When submitting your revision, we need you to address these additional requirements. 1. Please ensure that your manuscript meets PLOS ONE's style requirements, including those for file naming. The PLOS ONE style templates can be found at  https://journals.plos.org/plosone/s/file?id=wjVg/PLOSOne_formatting_sample_main_body.pdf and https://journals.plos.org/plosone/s/file?id=ba62/PLOSOne_formatting_sample_title_authors_affiliations.pdf 2. Please expand the acronym “NIH/NINDS, NIH/NEI” (as indicated in your financial disclosure) so that it states the name of your funders in full. This information should be included in your cover letter; we will change the online submission form on your behalf. 3. Please note that funding information should not appear in the Acknowledgments section or other areas of your manuscript. We will only publish funding information present in the Funding Statement section of the online submission form. Please remove any funding-related text from the manuscript.  4. When completing the data availability statement of the submission form, you indicated that you will make your data available on acceptance. We strongly recommend all authors decide on a data sharing plan before acceptance, as the process can be lengthy and hold up publication timelines. Please note that, though access restrictions are acceptable now, your entire data will need to be made freely accessible if your manuscript is accepted for publication. This policy applies to all data except where public deposition would breach compliance with the protocol approved by your research ethics board. If you are unable to adhere to our open data policy, please kindly revise your statement to explain your reasoning and we will seek the editor's input on an exemption. Please be assured that, once you have provided your new statement, the assessment of your exemption will not hold up the peer review process. 5. PLOS ONE now requires that authors provide the original uncropped and unadjusted images underlying all blot or gel results reported in a submission’s figures or Supporting Information files. This policy and the journal’s other requirements for blot/gel reporting and figure preparation are described in detail at https://journals.plos.org/plosone/s/figures#loc-blot-and-gel-reporting-requirements and https://journals.plos.org/plosone/s/figures#loc-preparing-figures-from-image-files. When you submit your revised manuscript, please ensure that your figures adhere fully to these guidelines and provide the original underlying images for all blot or gel data reported in your submission. See the following link for instructions on providing the original image data: https://journals.plos.org/plosone/s/figures#loc-original-images-for-blots-and-gels.    In your cover letter, please note whether your blot/gel image data are in Supporting Information or posted at a public data repository, provide the repository URL if relevant, and provide specific details as to which raw blot/gel images, if any, are not available. Email us at plosone@plos.org if you have any questions.

Reviewers' comments:

Reviewer's Responses to Questions

**Comments to the Author**

1. Is the manuscript technically sound, and do the data support the conclusions?

Reviewer #1: Yes

2. Has the statistical analysis been performed appropriately and rigorously?

Reviewer #1: Yes

3. Have the authors made all data underlying the findings in their manuscript fully available?

Reviewer #1: No

4. Is the manuscript presented in an intelligible fashion and written in standard English?

Reviewer #1: Yes

Reviewer #1: This manuscript by Hanes et al. reports the generation of a discrete Pcdhgc5 loss-of-function mouse model and examines its impact on cortical neuronal organization, survival, dendritic arborization, synaptic protein expression, and transcriptomic profiles. The study provides valuable insights into the potential roles of γC5 in cortical dendrite development. The experiments are clearly presented, and the manuscript is well written. Here are my two minor concerns. 1. The RNA-seq analysis highlights alterations in pathways related to synaptic activity, learning/memory, and circadian regulation. While this is interesting, no functional assays (e.g., behavioral tests or electrophysiology) were included that might link these transcriptomic observations to physiological outcomes. 2. The transcriptomic profiling was conducted at a single postnatal timepoint, which may not reflect earlier, transient developmental effects, especially given the known timing of γC5 expression onset. Additionally, several differentially expressed genes were not independently validated. Clarifying these points or adding a brief discussion would strengthen the conclusion.

**Do you want your identity to be public for this peer review?** For information about this choice, including consent withdrawal, please see our Privacy Policy

Reviewer #1: No

---

## [Author Response · Author response to Decision Letter 1]

28 Jan 2026

We thank the editor and reviewer for their comments on the manuscript.

The reviewer raised two points:

“Here are my two minor concerns.

1. The RNA-seq analysis highlights alterations in pathways related to synaptic activity, learning/memory, and circadian regulation. While this is interesting, no functional assays (e.g., behavioral tests or electrophysiology) were included that might link these transcriptomic observations to physiological outcomes.

2. The transcriptomic profiling was conducted at a single postnatal timepoint, which may not reflect earlier, transient developmental effects, especially given the known timing of γC5 expression onset. Additionally, several differentially expressed genes were not independently validated.

Clarifying these points or adding a brief discussion would strengthen the conclusion.”

We have addressed these points by adding text to the discussion. We agree that the RNA-seq analysis would be bolstered by future studies with functional analyses and finer validation, including a time course to capture developmental effects. However, these analyses are outside the scope of the current study. Our main goals here were to describe the new mouse mutant targeting the �C5 isoform and to test this isoform’s role in dendrite arborization. As our findings conflict with a recent study that found deficient dendrite arborization in a different mouse purporting to target �C5, we put emphasis on the expression of other clustered protocadherin isoforms, and the RNA-seq data were an important component of this.

Other updates include formatting changes requested by the editor (e.g., removing funding information from the acknowledgements), adding the GEO accession information for the RNA-seq data, and minor corrections to Figure 2 (addition of a label to 2F, correction of asterisks in 2B). We have also included raw Western blots and an excel file with every datapoint.

---

## [Decision Letter · Decision Letter 1]

27 Feb 2026

A new mouse mutant with a discrete mutation in Pcdhgc5 reveals that the Protocadherin γC5 isoform is not essential for dendrite arborization in the cerebral cortex

PONE-D-25-60828R1

Dear Dr. Garrett,

We’re pleased to inform you that your manuscript has been judged scientifically suitable for publication and will be formally accepted for publication once it meets all outstanding technical requirements.

Kind regards,

Xiangming Zha, Ph.D.

Academic Editor

PLOS One

Additional Editor Comments (optional):

Reviewers' comments:

Reviewer's Responses to Questions

**Comments to the Author**

Reviewer #1: (No Response)

2. Is the manuscript technically sound, and do the data support the conclusions?

Reviewer #1: (No Response)

3. Has the statistical analysis been performed appropriately and rigorously?

Reviewer #1: (No Response)

4. Have the authors made all data underlying the findings in their manuscript fully available?

Reviewer #1: (No Response)

5. Is the manuscript presented in an intelligible fashion and written in standard English?

Reviewer #1: (No Response)

Reviewer #1: (No Response)

**Do you want your identity to be public for this peer review?** For information about this choice, including consent withdrawal, please see our Privacy Policy

Reviewer #1: No

---

## [Editor Report · Acceptance letter]

PONE-D-25-60828R1

PLOS One

Dear Dr. Garrett,

I'm pleased to inform you that your manuscript has been deemed suitable for publication in PLOS One. Congratulations! Your manuscript is now being handed over to our production team.

Kind regards,

on behalf of

Dr. Xiangming Zha

Academic Editor

PLOS One